# Zero-Overhead Introspection for Adaptive Test-Time Compute

**Rohin Manvi**[†§*]  **Joey Hong**[†]  **Tim Seyde**[‡§]

**Maxime Labonne**[§]  **Mathias Lechner**[‡§]  **Sergey Levine**[†]

## Abstract

Large language models excel at reasoning but lack key aspects of *introspection*, including the ability to anticipate their own success and the computation required to achieve it. Humans use real-time introspection to decide how much effort to invest, when to make multiple attempts, when to stop, and when to signal success or failure. Without this ability, LLMs struggle to make intelligent meta-cognition decisions. Test-time scaling methods such as Best-of-N drive up cost and latency by using a fixed budget of samples regardless of the marginal benefit of each one at any point in generation, and the absence of confidence signals can mislead people, prevent appropriate escalation to better tools, and undermine trustworthiness. Learned verifiers or reward models can provide confidence estimates, but do not enable adaptive inference and add substantial inference cost by requiring extra models or forward passes. We present ZIP-RC, which equips models with zero-overhead introspective predictions of reward and cost. At every token during generation, ZIP-RC reuses reserved or unused logits in the same forward pass as next-token prediction to output a joint distribution over final reward and remaining length—no extra models, architecture change, or inference overhead. This full joint distribution is used to compute a sampling utility which is the linear combination of the expected maximum reward, total compute, and latency of set of samples if generated to completion. During inference, we maximize this utility with meta-actions that determine which prefix of tokens to continue or initiate sampling from. On mixed-difficulty mathematical benchmarks, ZIP-RC improves accuracy by up to $12\%$ over majority voting at equal or lower average cost, and traces smooth Pareto frontiers between quality, compute, and latency. By providing real-time reward–cost introspection, ZIP-RC allows models to reason adaptively and more efficiently.

## 1 Introduction

The rapid evolution of large language models (LLMs) has enabled unprecedented capabilities in complex tasks ranging from general question-answering to automated coding and mathematical reasoning (Brown et al., 2020; Kojima et al., 2022; Wei et al., 2022). To become truly reliable, however, LLMs must develop a capacity for *introspection*: the ability to assess their own progress and anticipate the effort required to succeed. Humans can be introspective and can effectively act upon this information to make better decisions. If a model could predict its future success (reward) *and* the resources needed to achieve it (cost), it could allocate compute more effectively, expose likely failure modes before they occur, and provide transparent signals about confidence and anticipated "thinking time." A key obstacle has been that such introspection typically requires auxiliary mechanisms that add nontrivial computational overhead and complexity.

The need for introspection is growing more urgent as reasoning traces continue to lengthen. Recent work shows that scaling *test-time* compute through reasoning often yields larger performance gains

---

*Corresponding author, `rohinm@berkeley.edu`

[†]UC Berkeley

[‡]MIT CSAIL

[§]Liquid AI

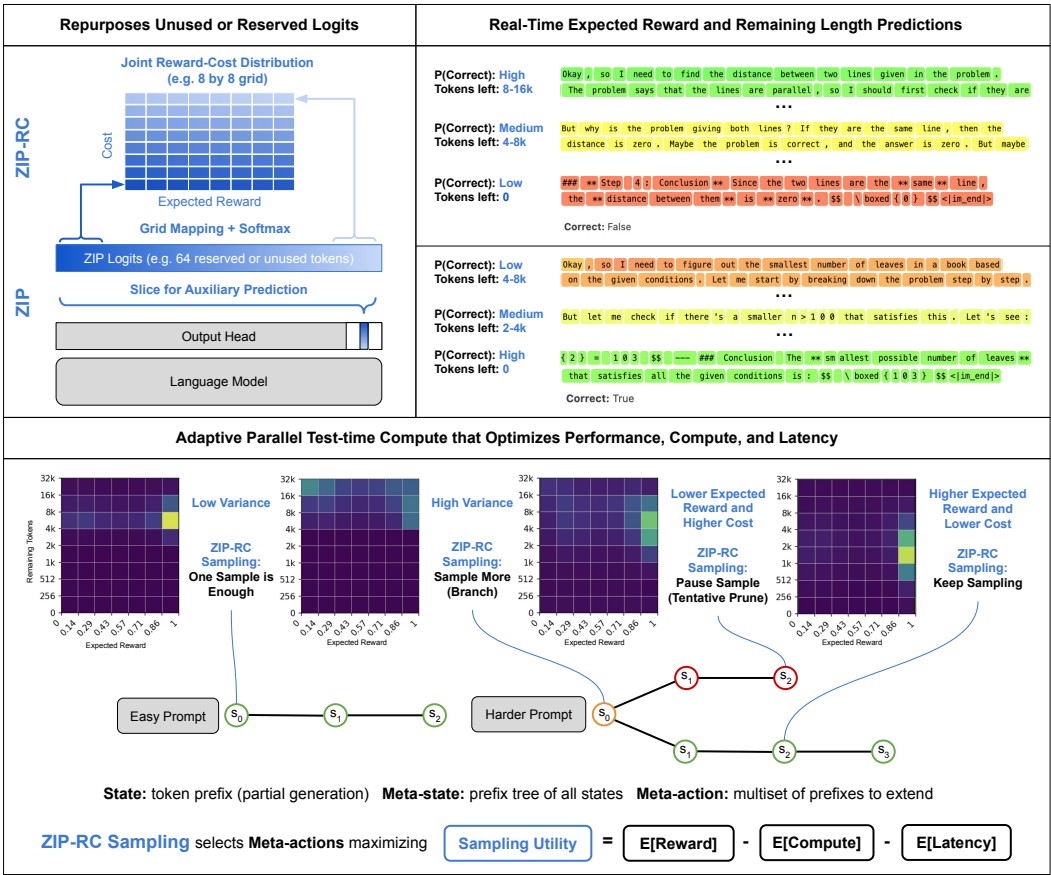

Figure 1: Top left shows how ZIP repurposes reserved or unused logits in the output head of a language model to instantiate auxiliary predictions, such as the grid mapping for the joint reward-cost distribution that ZIP-RC uses. Top right demonstrates how ZIP-RC can provide real-time expected reward and remaining length predictions. Finally, the bottom shows the joint distributions from ZIP-RC and how they indicate optimal sampling strategies. ZIP-RC sampling uses these joint distributions to calculate a sampling utility to autonomously select meta-actions for optimal test-time compute allocation.

than simply increasing model size (Wang et al., 2023b; Yao et al., 2023; Jaech et al., 2024; Snell et al., 2024; Guo et al., 2025). But performance has scaled only logarithmically with additional computation, forcing models to produce ever longer chains of thought—sometimes tens of thousands of tokens today and plausibly orders of magnitude more in the future (Wu et al., 2024). With time as a fundamental limiting resource, a critical question is how to use a fixed wall-clock budget to achieve the highest performance possible.

A promising approach is the canonical test-time scaling method *Best-of-N* (BoN) sampling, which generates $N$ candidates and selects the best using a learned verifier, reward model, or majority vote (Cobbe et al., 2021; Zheng et al., 2023; Kwon et al., 2023; Lightman et al., 2023b; Wang et al., 2023b). While appealing in theory due to its parallelism, BoN is not adaptive: every trajectory is carried to completion regardless of promise. On easy tasks this wastes computation, and on hard tasks it inflates latency, since wall-clock time is governed by the longest generation and both length and total compute grow with $N$ (Leviathan et al., 2023). What is missing is a way for models to anticipate which samples are worth continuing and which should be paused or abandoned, so that parallel effort is concentrated on trajectories most likely to succeed and fastest to complete.

Early-stopping and pruning methods aim to reduce BoN's inefficiency by terminating unpromising samples mid-generation (Fu et al., 2025; Huang et al., 2025). These approaches are valuable first steps toward adaptivity, but they typically rely on *scalar* signals—such as a confidence score from a classifier—or on simple heuristics. This creates two limitations. First, a scalar cannot capture

the central reward–cost trade-off: a low-confidence trajectory may be worthwhile if nearly finished, while a high-confidence one may be impractical if it implies a long, costly continuation. Second, these methods do not quantify the marginal benefit of drawing more samples, which depends on the entire reward distribution rather than its expectation. As a result, such strategies can reduce compute in some cases but often fail to improve wall-clock time, falling short of the broader goal of enabling models to allocate compute adaptively—expending more effort on difficult queries and less on easy ones (Manvi et al., 2024; Graves, 2016).

We introduce ZIP-RC, which addresses these limitations by training language models to provide zero-overhead introspective predictions of the *joint distribution* over reward and cost. At each decoding step, unused vocabulary logits parameterize a joint distribution over final reward and remaining generation length (see fig. 1). Access to the full joint—not just a scalar—enables order-statistic calculations that quantify the marginal utility of continuing partial samples or spawning additional samples. For example, when the predicted reward distribution has high variance, allocating more samples can substantially increase the expected maximum reward. We maximize a *sampling utility* that explicitly balances accuracy, compute, and latency through a linear combination of their expectations. The coefficients of the linear combination can be tuned to the desired balance of reward, compute, and latency. Optimizing this utility produces the behaviors observed in our experiments: when latency is prioritized, ZIP-RC spawns larger pools of samples and schedules early pruning to chase an early finisher; when compute is prioritized, it deprioritizes low-value trajectories aggressively and allocates more samples only when they are likely to pay off.

Experiments on mixed-difficulty mathematical benchmarks show that ZIP-RC improves accuracy by up to $12\%$ over majority voting while using less average cost. By adjusting the utility coefficients, it traces smooth Pareto frontiers between accuracy, compute, and latency. We contribute ZIP and ZIP-RC which enable models to be introspective for more interpretable generations and maximize a sampling utility to improve performance with fixed compute and latency.

## 2 RELATED WORK

Improving the efficiency and reliability of LLM reasoning requires both new methods for guiding generation and principled strategies for allocating computational resources at inference time. Our work builds on three key areas of research: the use of verifiers for response selection, process-level rewards for fine-grained feedback, and adaptive inference strategies for efficient computation.

**Verifiers and reward models for output selection.** A common approach to enhancing LLM performance is to train an external verifier or reward model (RM) to assess the quality of complete responses. Such models provide outcome-based feedback, typically assigning a scalar score or probability of correctness to an entire output sequence. Outcome RMs have been widely used in reasoning and alignment works, from math problem solving to preference-based fine-tuning (Cobbe et al., 2021; Yu et al., 2023; Stiennon et al., 2020). They can be integrated during training, as in reinforcement learning settings (Ouyang et al., 2022; Bai et al., 2022), or applied at inference time through selection strategies such as Best-of-N sampling (Cobbe et al., 2021; Li et al., 2022). Recent work has explored unifying the generator and verifier, using the model's own logits for certain tokens as a proxy for a reward model (Ren et al., 2023). Our work extends this introspective direction, moving beyond scalar correctness prediction to modeling a joint distribution over the expected future reward and computational cost at every token.

**Process-based rewards for fine-grained feedback.** A limitation of outcome-supervision is its reliance on a sparse reward signal that makes credit assignment challenging, especially for long reasoning chains. Process-based reward models (PRMs) instead score intermediate steps via human annotation (Lightman et al., 2023b), LLM-as-judge (Zheng et al., 2023), or automated token-level value estimates. These automated estimates can be generated by propagating final outcome rewards back to individual tokens (Liu et al., 2024) or through other value estimation techniques (Uesato et al., 2022; Luo et al., 2024). While most PRMs aim to improve the training signal, our goal is distinct: we use predictive feedback in real time to guide inference itself. Closest to the calibration side of this literature, Damani et al. (2025) augment a binary correctness reward with a confidence score to improve model calibration. Our approach is complementary: rather than training for calibrated confidence, we predict a joint distribution over future reward and future cost, turning process-level signals into a direct control knob for utility-aware inference.

**Adaptive inference and introspective models.** Our work enables a form of adaptive inference, a long-standing goal in machine learning (Graves, 2016; Bengio et al., 2015) that has become increasingly critical for large models (Snell et al., 2024). Adaptive methods that use multiple models or sequential sampling have been explored (Damani et al., 2024; Wang et al., 2024). A more recent direction has involved parallel sampling that includes the pruning of unpromising generation paths. For instance, recent methods terminate samples based on mid-generation confidence scores(Manvi et al., 2024; Fu et al., 2025) or prune exploration based on step-wise consistency checks (Aggarwal et al., 2023). We advance this line of work with a more general formulation: instead of relying on simple heuristics for pruning, we use our joint reward-cost predictions to explicitly optimize a utility function. This enables a richer set of meta-actions, such as dynamically resizing the sample pool and reallocating budget across trajectories. Conceptually, our approach parallels the integration of value functions with search in reinforcement learning (Silver et al., 2016), where predictive signals guide exploration. It is also complementary to inference optimization techniques like speculative decoding (Leviathan et al., 2023), which accelerate generation at the token level. By providing real-time estimates of success and cost, the predictions from ZIP-RC contribute to a broader vision of introspective models that report their internal states (Binder et al., 2024; Kadavath et al., 2022), enhancing efficiency and interpretability.

## 3 PRELIMINARIES

**Generation as a token-level MDP.** We formalize text generation as a finite-horizon Markov Decision Process (MDP), following Ramamurthy et al. (2022). The MDP is defined by the tuple $M = (\mathcal{S}, \mathcal{A}, R, P, \gamma, H)$ over a finite vocabulary $\mathcal{V}$, where $\mathcal{S}$ is the state space, and $\mathcal{A}$ the action space, $R$ the reward function, $P$ the transition function, $\gamma \in [0, 1]$ the discount factor, and $H$ the horizon. Given an input prompt $\mathbf{x} = (x_0, \ldots, x_m)$ consisting of tokens in the vocabulary $x_i \in \mathcal{V}$, the initial state is $s_0 = \mathbf{x}$. At timestep $t$, the LLM acts as a policy $\pi(a_t|s_t)$ that outputs the probability distribution over actions $a_t \in \mathcal{V}$. The transition function $P$ deterministically appends $a_t$ to state $s_t$, yielding next state $s_{t+1} = (x_0, \ldots, x_m, a_0, \ldots, a_t)$. The episode terminates when the model emits an end-of-sequence token <EOS> or the length of the generated sequence reaches the horizon $H$. Upon termination at timestep $T$, the environment returns a terminal reward $R(s_T)$. The discount factor is defined as $\gamma = 1$ and the value of any state $s_t$ under policy $\pi$ is the expected terminal reward from that state onward $V(s_t) = \mathbb{E}_\pi [R(s_T) \mid s_t]$.

**Best-of-N.** Best-of-$N$ (BoN) is an inference-time selection mechanism that decouples generation from evaluation to improve output quality. Given a prompt $\mathbf{x}$ and a generator policy $\pi$, the method draws $N$ independent and identically distributed (i.i.d.) terminated states $s_T^{(1)}, \ldots, s_T^{(N)}$ from the policy. A learned verifier $\hat{V} : \mathcal{V}^* \to \mathbb{R}$, typically a reward model, then assigns a scalar score to each terminated state. The final output is the state with the highest score, selected as

$$s_T^* \in \arg\max_{i \in [N]} \hat{V}(s_T^{(i)}). \tag{1}$$

The selection depends only on the relative ordering of scores from $\hat{V}(\cdot)$, ties are broken arbitrarily.

## 4 ZERO-OVERHEAD INTROSPECTIVE PREDICTION OF REWARD & COST

We introduce Zero-overhead Introspective Prediction (ZIP), a method for extracting auxiliary signals during inference without extra models, architectural changes, or forward passes. ZIP repurposes the logits of a small set of reserved tokens to parameterize these auxiliary predictions within the same forward pass that generates the next-token probabilities. We then instantiate ZIP for reward and cost prediction (ZIP-RC).

**Zero-overhead introspective prediction (ZIP).** Let $\mathcal{V}$ be the vocabulary and $\mathcal{R} \subset \mathcal{V}$ a fixed contiguous set of *reserved tokens*. At decoding step $t$, the model produces logits $z_t \in \mathbb{R}^{|\mathcal{V}|}$. ZIP interprets logits over $\mathcal{R}$ as parameters of an auxiliary predictor (e.g. via a softmax). A rough visualization of this is shown in the top right of fig. 1. Before sampling, these logits are masked to

remove probability mass:

$$\pi_\theta(a_t \mid s_t) = \begin{cases} \dfrac{\exp(z_t[a_t])}{\sum_{v \in \mathcal{V} \setminus \mathcal{R}} \exp(z_t[v])}, & a_t \in \mathcal{V} \setminus \mathcal{R}, \\ 0, & a_t \in \mathcal{R}. \end{cases} \tag{2}$$

Thus, each forward pass yields both (i) the decoding distribution on $\mathcal{V} \setminus \mathcal{R}$ and (ii) auxiliary predictions from $z_t[\mathcal{R}]$, incurring **zero additional cost** at inference time.

During training, we supervise the auxiliary head via a task-specific loss $\mathcal{L}_{\text{aux}}$ applied to $z_t[\mathcal{R}]$ (e.g., cross-entropy for categorical targets, Bernoulli NLL for binary targets, MSE for continuous targets), while regularizing the policy toward a frozen copy of the original policy $\pi$:

$$\mathcal{L}(s_t) = \mathcal{L}_{\text{aux}}(s_t) + \alpha_{\text{KL}} \, \text{KL}(\pi_\theta(\cdot \mid s_t) \, \| \, \pi(\cdot \mid s_t)). \tag{3}$$

ZIP is agnostic to the prediction target or loss, it simply standardizes how auxiliary predictions are produced during inference, with zero inference overhead. An alternative that keeps the model frozen is discussed in  section A.5.

**ZIP-RC: joint reward-cost distribution prediction.** We use ZIP to predict a *joint distribution* over the (expected) reward and remaining length of a rollout using $\pi$ starting from any prefix $s_t$. We define the random variables

$$Z^\pi(s_t) = \mathbb{E}\left[R(s_T)\right], \qquad L^\pi(s_t) = |s_T| - |s_t|, \tag{4}$$

for the expected terminal reward and length. In practice, we approximate the expected terminal reward with $\hat{V}(s_T)$ from a learned verifier. We discretize the range of values into $B_V$ bins with boundaries $\{v_b\}_{b=1}^{B_V+1}$ and lengths into $B_T$ bins with boundaries $\{t_\ell\}_{\ell=1}^{B_T+1}$, assigning one reserved token per $(b, \ell)$ using index in the output vocabulary $\mathcal{V}$ given by $i_{b,\ell} = i_\mathcal{R} + (b-1)B_T + (\ell-1)$, where $i_\mathcal{R}$ is the index of the first reserved token. Let $z_t^{\text{aux}}(b, \ell) \equiv z_t[r_{b,\ell}]$. The joint distribution is

$$p_\theta(b, \ell \mid s_t) = \frac{\exp(z_t^{\text{aux}}(b, \ell))}{\sum_{b'=1}^{B_V} \sum_{\ell'=1}^{B_T} \exp(z_t^{\text{aux}}(b', \ell'))}. \tag{5}$$

A rough visual representation of the grid mapping and examples of this learned distribution is shown in  fig. 1. Given a completed trajectory $s_T$, for each timestep we construct training targets for each prefix $s_t$ by computing $(b^*, \ell^*)$ such that

$$\hat{V}(s_T) \in [v_{b^*}, v_{b^*+1}), \qquad |s_T| - |s_t| \in [t_{\ell^*}, t_{\ell^*+1}). \tag{6}$$

Finally, we train with cross-entropy $\mathcal{L}_{\text{aux}}(s_t) = -\log p_\theta(b^*, \ell^* \mid s_t)$, together with the policy-preserving KL above. Other practical implementation details are discussed in  section A.3.

**Why expected reward instead of realized reward?** It may initially seem unnatural to use an estimated value $\hat{V}(s_T)$ by a trained critic rather than the realized reward. To explain this choice, let $s_T^{(1)}, \ldots, s_T^{(N)} \overset{\text{i.i.d.}}{\sim} \pi_\theta(\cdot \mid s_0)$ be completions for a prompt, and $\hat{V}$ the estimated value function used in BoN selection. The chosen index $s_T^* = \arg\max_i \hat{V}(s_T^{(i)})$ yields score $\hat{V}(s_T^*) = \max_{i \in [N]} \hat{V}(s_T^{(i)})$. Modeling the distribution of possible terminal values $V^\pi(s_T)$ via $\hat{V}(s_T)$ rather than possible terminal rewards $R(s_T)$: (i) aligns with the actual selection objective, and (ii) admits closed-form order-statistic expectations since noisy environment rewards from $R(s_T)$ cannot be assumed to be independent but its expectation $V(s_T)$ can.

**ZIP-RC for sample selection and interpretability.** Using the learned joint distribution, we can also compute individual marginal distributions:

$$q_\theta^V(b \mid s_t) = \sum_{\ell=1}^{B_T} p_\theta(b, \ell \mid s_t), \qquad q_\theta^L(\ell \mid s_t) = \sum_{b=1}^{B_V} p_\theta(b, \ell \mid s_t), \tag{7}$$

which can be used to estimate the value and the expected remaining tokens to completion

$$V^\pi(s_t) = \mathbb{E}[Z^\pi(s_t)] \approx \sum_{b=1}^{B_V} \frac{v_b + v_{b+1}}{2} \, q_\theta^V(b \mid s_t), \qquad \mathbb{E}[L^\pi(s_t)] \approx \sum_{\ell=1}^{B_T} \frac{t_\ell + t_{\ell+1}}{2} \, q_\theta^L(\ell \mid s_t). \tag{8}$$

Here, the value estimation can be used for final sample selection and both the value and the expected remaining tokens to completion act as confidence and "thinking time" signals. A rough visualization of the interpretable signals is shown in the top right of fig. 1.

## 5    TEST-TIME COMPUTE USING ZIP-RC (ZIP-RC SAMPLING)

While large language models are post-trained to maximize the likelihood of high-reward generations, they remain imperfect policies due to finite data and compute. Low-reward completions are often sampled even when their deficiencies are apparent—either implicitly through low likelihood or explicitly via external reward models. Even greedy decoding (temperature 0) does not guarantee high-likelihood or high-reward outputs. Thus, one-shot sampling is insufficient for reliably accomplishing tasks. Test-time methods such as majority voting, BoN, Weighted BoN, and Pass@k show the alternative: by actively searching across multiple trajectories, they substantially outperform single-sample decoding. The performance gap highlights that the gain comes from active *search*.

Existing test-time methods, however, are heuristic and often inefficient. BoN can, in principle, explore as much as all other approaches with a large enough $N$, but this is impractical given compute and latency constraints. While more sophisticated search strategies like beam search try to explore more efficiently by allowing for intermediate branching and pruning at intervals, one could imagine removing constraints on search further. Though the goal of test-time search is clear—maximize task success while minimizing generation cost—prior methods do not achieve this in a principled way. Our goal is to propose a method that does. In this section, we introduce *ZIP-RC sampling*, which leverages predictions from ZIP-RC to explicitly optimize generations for both success and cost. We provide a high-level overview of the framework here (visually summarized in fig. 1) and provide the full formalisms and derivations in section A.1 and section A.2.

**Test-time compute as Decision-Making Under a Meta-MDP.**    We formalize the problem of test-time compute as decision-making under a high-level *meta-MDP*, detailed in section A.1. The state of this MDP is the current prefix tree (the set of all partial generations). At each step, the "meta-action" determines which prefixes in the tree to extend or branch from. Prefixes that are not selected are effectively *paused* rather than discarded. The objective is to maximize a meta-reward defined as the final correctness of the best answer minus the generation cost incurred. Crucially, this cost function includes both *total compute* (sum of tokens generated) and *latency* (depth of the longest trajectory), balanced by coefficients $\alpha$ and $\beta$. This formalism allows us to treat inference not as a static procedure, but as a dynamic resource allocation problem.

**The Sampling Utility.**    Solving for the optimal policy in this meta-MDP is intractable. Instead, we approximate the optimal value function using a quantity we call the *sampling utility*. As derived in section A.2, the sampling utility estimates the value of a specific, interpretable strategy: performing rollouts from the current set of candidates, but with the capability to pause them at optimized future horizons. Maximizing this utility allows the controller to explicitly balance the marginal benefit of adding more samples (higher probability of finding a high-reward answer) against the marginal cost of computation and time. This sampling utility can be computed tractably using the joint predictions described in Section 4. Because ZIP-RC predicts the *joint distribution* of reward and remaining length, we can compute required order statistics—such as the expected maximum reward of a set of samples or the expected latency given a specific pausing schedule—in closed form. Note this requires lightweight CPU-based calculations that are negligible compared to the forward pass of an LLM.

**Sampling Loop**    At inference time, ZIP-RC sampling operates as a meta-policy. At regular intervals, it evaluates the sampling utility of various candidate meta-actions (e.g., pausing weak samples, branching strong ones, or continuing the current set). It selects the action that maximizes this utility (as visualized in the bottom panel of fig. 1) and executes it for the next decoding steps. This allows the model to adapt online: if trajectories are projected to be low-value or excessively costly, the system redirects computation elsewhere. We discuss practical implementation details, such as normalizing cost coefficients and reducing the search space, in section A.4.

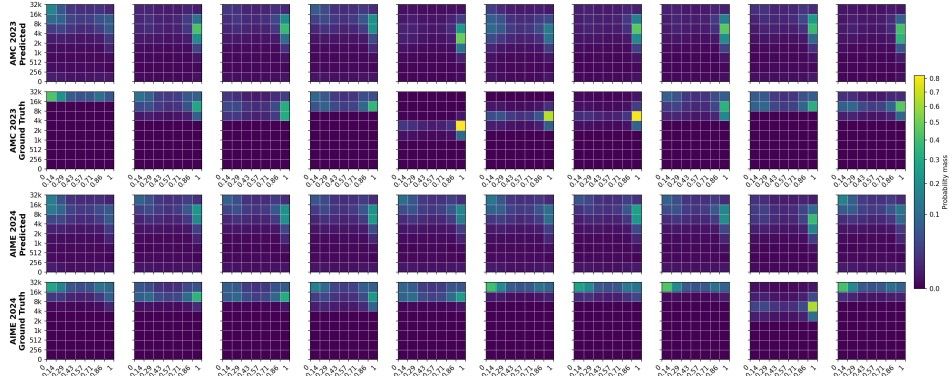

Figure 2: Predictions and ground truth for the initial joint distributions of 10 questions randomly sampled from the AMC 2023 benchmark and 10 questions from the AIME 2024 benchmark. The ground truth for each prompt was estimated with 256 rollouts from Qwen3-1.7B, and predictions were made using ZIP-RC trained with the same model. This shows that the joint distribution from ZIP-RC is calibrated and relatively accurate in forecasting the outcomes of its own rollouts.

| Model | Beginning (Reward+Cost) | End (Reward) | | |
| | Total Variation | F1 Score | Accuracy | Recall (Incorrect) |
| --- | --- | --- | --- | --- |
| Qwen3-1.7B | 0.46 | 0.91 | 0.88 | 0.82 |
| LFM2-1.2B | 0.45 | 0.91 | 0.87 | 0.69 |
| LFM2-350M | 0.48 | 0.80 | 0.82 | 0.87 |

Table 1: Prediction accuracy of ZIP-RC at the beginning and end of generation. At the beginning, no ground-truth reward or remaining-length label exists due to stochastic decoding, so we evaluate the joint reward-cost prediction using Total Variation. At the end of generation, the ground-truth reward is known, allowing us to report F1 score, accuracy, and incorrect-answer recall using a threshold of 0.5.

## 6 EXPERIMENTS

Our experiments aim to test the following hypotheses:

(1) ZIP-RC can accurately predict the joint reward-cost distribution.
(2) ZIP-RC sampling can be tuned to balance between output quality, and compute cost and latency, tracing a Pareto frontier over the quantities over strong inference baselines.
(3) ZIP-RC sampling is adaptive and generalizes across tasks of varying difficulty and across models of varying size.

We will describe and present results that provide positive evidence for each hypothesis individually.

### 6.1 EXPERIMENTAL SETUP

**Models.** We use three open models spanning capability and scale: *Qwen3-1.7B* (Alibaba) in reasoning mode (Yang et al., 2025); *LFM2-1.2B Math* (Liquid AI), a compact mathematical-reasoning model (LiquidAI, 2025); and *LFM2-350M Math*, a smaller variant targeting efficient math reasoning. Unless stated otherwise, decoding is identical across methods; ZIP-RC modifies only the sampling policy at inference time.

**Training data for ZIP-RC and baselines.** We construct a mathematical training corpus by combining DeepScaleR (Luo et al., 2025), the MATH training split (Hendrycks et al., 2021), and the GSM8K training split (Cobbe et al., 2021). For each prompt, we generate two on-policy rollouts per model, yielding roughly 100k rollouts in total. We then label each rollout for correctness against the ground-truth answer. These labeled rollouts are used to train model-specific ZIP-RC predictors as well as any learned baselines.

**Baselines.** We evaluate against the following baselines that consist of popular sampling strategies that fall under the parallel sampling paradigm where multiple candidate samples are generated in parallel and there is some selection method. Other notable paradigms include beam search or self-

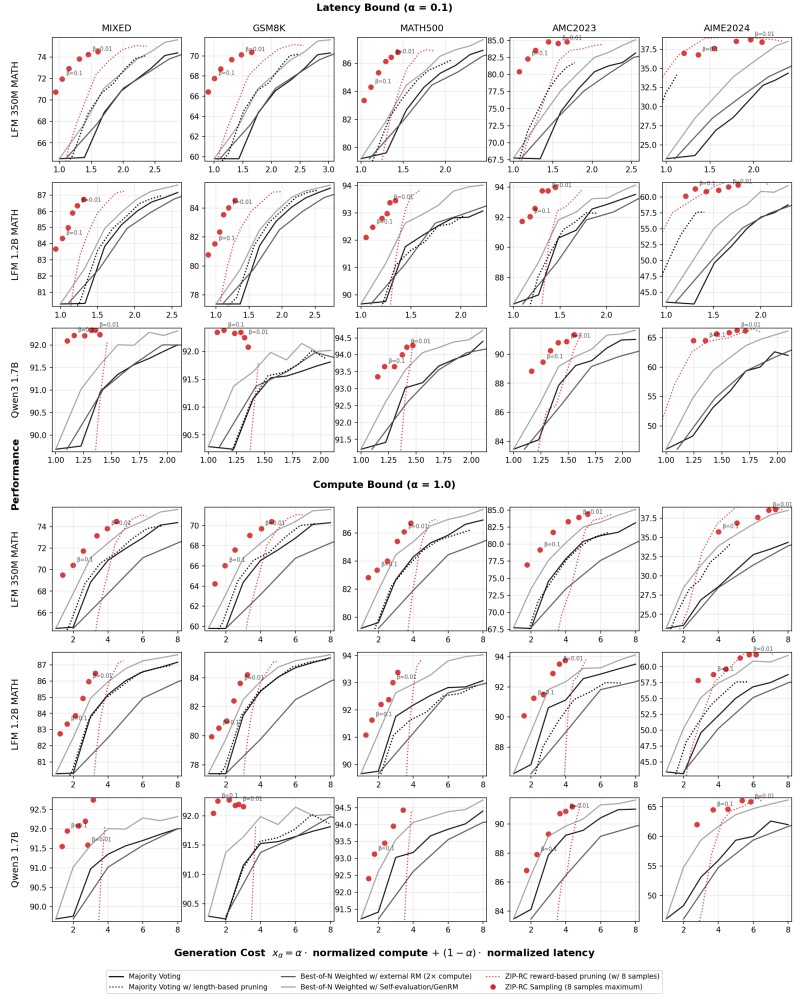

Figure 3: Performance of ZIP-RC sampling and baselines across all models and benchmarks. The top half demonstrates the latency bound setting where $\alpha = 0.1$, and the bottom half demonstrates the compute bound setting where $\alpha = 1.0$. Adjusting $\beta$ in ZIP-RC sampling allows it to trade generation cost for higher performance (similar to increasing $N$ in BoN) while adjusting $\alpha$ allows it to adjust the prioritization of compute and latency. By navigating the Pareto frontier and allocating compute adaptively, ZIP-RC sampling significantly outperforms majority voting and other baselines.

refinement. However, we use parallel sampling methods, which are the most commonly used and reported as they do not suffer from collapsing diversity issues that arise from branching and generating with similar prefixes or the ballooning latency issues from methods that generate samples sequentially. We use stronger adaptations of Best-of-$N$ (BoN), and an ablation of ZIP-RC that performs pruning without the sampling utility optimization and instead uses the expected reward directly.:

(1) *Majority Voting (MV)* (self-consistency), which selects the most frequent final answer, breaking ties uniformly at random (Wang et al., 2023a). This is an extremely common method since it does not require any learned verifier.

(2) *MV with length-based pruning*, which discards very long, potentially looping samples (cut at 8k tokens). This baseline acts as a sanity check to see if our latency gains only come from preventing looping samples from generating to the maximum 32k generation length.

(3) *Weighted BoN with external RM*, which scores each sample with a separate reward model trained on the same math corpus; because the RM reprocesses the full sequence without KV cache, FLOPs roughly double relative to generation alone (Li et al., 2023). This baseline demonstrates strong performance that goes beyond Best-of-N sampling.

| Model | Method | Gen. Cost | AIME2024 | AMC2023 | MATH-500 | GSM8K | Mixed |
|-------|--------|-----------|----------|---------|----------|-------|-------|
| Qwen3-1.7B | ZIP-RC sampling | 1.43 | 65.8 | 90.9 | 94.1 | 92.2 | 92.2 |
| | Majority Voting | 1.40 | 53.1 | 87.9 | 93.0 | 91.2 | 91.0 |
| | MV length-prune | 1.46 | 25.1 | 58.5 | 84.7 | 91.6 | 88.0 |
| | Weighted BoN ext. RM | 1.43 | 54.7 | 86.5 | 92.6 | 91.4 | 91.0 |
| | Weighted BoN Self-eval | 1.40 | 59.4 | 89.1 | 93.6 | 91.6 | 91.6 |
| | ZIP-RC reward prune | 1.33 | 43.3 | 86.0 | 90.3 | 89.6 | 88.9 |
| LFM2-1.2B | ZIP-RC sampling | 1.35 | 60.9 | 93.5 | 93.4 | 83.6 | 86.0 |
| | Majority Voting | 1.60 | 49.6 | 90.6 | 91.8 | 81.4 | 83.8 |
| | MV length-prune | 1.70 | 51.3 | 89.8 | 91.6 | 83.0 | 84.9 |
| | Weighted BoN ext. RM | 1.53 | 50.3 | 89.0 | 91.1 | 79.8 | 82.5 |
| | Weighted BoN Self-eval | 1.60 | 55.1 | 91.8 | 92.6 | 82.5 | 84.9 |
| | ZIP-RC reward prune | 1.49 | 57.5 | 90.2 | 92.5 | 83.8 | 85.8 |
| LFM2-350M | ZIP-RC sampling | 1.49 | 38.8 | 83.9 | 86.1 | 70.1 | 74.1 |
| | Majority Voting | 1.70 | 26.9 | 74.5 | 82.7 | 64.4 | 68.8 |
| | MV length-prune | 1.66 | 28.3 | 74.8 | 83.6 | 66.5 | 70.6 |
| | Weighted BoN ext. RM | 1.59 | 28.5 | 73.4 | 81.9 | 63.2 | 67.8 |
| | Weighted BoN Self-eval | 1.70 | 31.4 | 77.6 | 84.4 | 66.8 | 71.1 |
| | ZIP-RC reward prune | 1.27 | 21.7 | 69.7 | 83.2 | 63.0 | 67.8 |

Table 2: Performance and generation cost at $\alpha = 0.1$ under matched-cost configurations. ZIP-RC sampling uses $\beta = 0.01$ and a maximum of eight samples. MV uses three samples; MV length-prune uses four; Weighted BoN Self-eval (GenRM) uses three; Weighted BoN with external RM uses two; and ZIP-RC reward prune uses a $0.4$ threshold with eight samples.

(4) *Weighted BoN with self-evaluation (GenRM)*, which replaces the external RM with trained self-evaluations derived from the generator (Manvi et al., 2024; Zhang et al., 2025; Mahan et al., 2024). We specifically include this baseline as it is another method that uses less compute than external reward models for selection.

(5) *ZIP-RC with reward-based pruning*, which starts with a fixed pool and prunes any trajectory whose predicted expected reward falls below a threshold using ZIP-RC's real-time signal. This acts as a natural and strong ablation to our sampling utility optimization as it directly prunes weak samples that have less promise than those with high expected reward.

**Benchmarks.** We report performance on *AIME 2024*, *AMC 2023*, *MATH-500* (Lightman et al., 2023a), and *GSM8K*. We additionally evaluate on a *Concatenated Mixed-Difficulty Benchmark* formed by concatenating the above, which probes adaptive allocation across difficulties.

**Metrics.** First and foremost we measure accuracy on each benchmark as it is an obvious and good measure for performance and high-quality responses. Beyond performance, we measure efficiency and latency. *Normalized compute* reports total FLOPs per prompt normalized by the FLOPs of a single-sample generation for that prompt. We compute FLOPs with the standard $2N$ rule (proportional to the sum of input and generated tokens) and account for KV caching where applicable. *Normalized best-case latency* measures the lower bound on wall-clock time as the maximum number of sequential forward passes across the candidate set; with unconstrained data-parallel sampling, latency is governed by the longest trajectory. *Generation cost* aggregates these via a linear combination, $\text{GenCost} = \alpha \cdot \text{NormCompute} + (1 - \alpha) \cdot \text{NormLatency}$. Unless otherwise specified, we use $\alpha = 0.1$, which roughly balances compute and latency in typical parallel regimes (e.g., eight parallel samples often behave like two to three serial generations in practice). For ZIP-RC sampling we sweep $\beta$, which trades off expected quality against cost in the utility; when reporting matched-cost comparisons we set $\beta = 0.005$ and cap the pool at $8$ samples for fair comparison to other baselines.

## 6.2 ACCURACY OF ZIP-RC'S REAL-TIME PREDICTIONS

ZIP provides auxiliary predictions with zero overhead, but for this to be useful they must be reliable. We first visually validate whether the joint reward-cost distribution predictions from ZIP-RC are reasonable. To do so, we first obtain ground truth estimates of the joint distributions at the start of generation on AMC 2023 + AIME 2024, which exhibit nontrivial error rates and diverse reasoning trace lengths. The ground truth estimates are derived with 256 rollouts from Qwen3-1.7B and the predictions are made using ZIP-RC trained with the same model. From the 10 random examples from each benchmark in fig. 2 we can see that the predictions are calibrated and relatively accurate in forecasting the distribution of outcomes.

To quantitatively validate the accuracy of the predictions, we measure the total variation at the beginning of generation using the same ground truth joint distribution estimates, as well as standard classification metrics for reward prediction using a threshold of $0.5$. As seen in table 1, the total variations from the ground truth confirm the visual validation that the predicted distributions are rel-

atively close to the ground truth, and the reward prediction at the end of generation further confirms this; it demonstrates high accuracy in terms of F1 Score, accuracy, and recall for incorrect answers (using a threshold of 0.5). Overall, these results indicate that ZIP-RC and ZIP predictions can be calibrated and accurate despite being done in the same forward pass as next-token prediction.

## 6.3 TRACING THE QUALITY–COMPUTE–LATENCY FRONTIER

We next test whether maximizing the sampling utility with specific cost coefficients achieves controllable tradeoffs. At each decision point, ZIP-RC evaluates meta-actions that serve three complementary purposes. First, initiating new samples only when necessary and avoiding continuing to generate low-value trajectories saves compute, which is reflected in the compute bound setting in the bottom half of fig. 3 ($\alpha = 1.0$), where ZIP-RC achieves compute savings. Second, penalizing the continued sampling of long outliers avoids samples that would dominate latency. Third, expanding the initial pool of samples while planning to use a near-term maximum horizon enables the search to pursue early finishers without paying the full wall-clock cost of long runs. These mechanisms together drive the latency savings observed in the top half of fig. 3 ($\alpha = 0.0$). In both settings $\beta$ is successfully used similar to $N$ in BoN in order to increase performance for more generation cost. Parameters $\alpha$ and $\beta$ together thus provide simple control knobs over compute–latency emphasis and quality–cost trade-off.

Across both $\alpha$ regimes, ZIP-RC sampling traces smooth Pareto frontiers that strictly dominate MV across benchmarks and scales validating that a single utility can jointly improve quality, compute, and latency. When $\alpha = 0.1$ (latency-emphasis), it substantially reduces cost, with the largest relative reduction observed on LFM2-350M (up to roughly 40%). Because we cap at eight samples, the frontier saturates once pass@8 performance is reached for a given $\beta$.

## 6.4 ADAPTIVE INFERENCE WITH ZIP-RC SAMPLING

Finally, we compare ZIP-RC sampling against all baselines at matched generation cost with $\alpha = 0.1$ in table 2. Two patterns emerge: (i) at fixed cost, ZIP-RC improves accuracy relative to MV and weighted BoN baselines; (ii) it allocates more samples to harder instances (AIME/AMC) and to weaker models, while pruning aggressively on easier problems or stronger models.

At matched cost, ZIP-RC sampling improves accuracy over MV and weighted BoN on all models and benchmarks. On harder subsets such as AIME 2024, gains reach up to 12% absolute while using less average cost. The adaptive policy naturally uses more samples when the predicted reward distribution is high-variance—where the expected benefit of best-of-N is greatest—and conserves compute when one trajectory is expected to be dominant. This pattern is evident on the mixed-difficulty benchmark (left-most column in fig. 3) and across model scales: weaker models and harder tasks receive more samples, leading to higher overall accuracy.

**Takeaways.** ZIP-RC's real-time predictions are accurate and reliable enough to enable principled search during decoding. This yields (i) reliable mid-generation detection of weak or overlong trajectories, (ii) smooth and tunable Pareto frontiers between quality, compute, and latency, and (iii) adaptive allocation that consistently outperforms fixed-budget Best-of-$N$ at the same or lower cost.

## 7 CONCLUSION

We introduced ZIP-RC, a zero-overhead framework for introspective inference that predicts future reward and cost by repurposing existing logits. This enables principled, real-time decoding search during inference, yielding up to 12% absolute accuracy gains over strong Best-of-N baselines at a lower average cost, while tracing a smooth Pareto frontier between quality, compute, and latency. These findings open natural extensions, such as applying it to diverse domains and testing fully dynamic resource allocation across different models and reasoning modes. Ultimately, ZIP-RC marks a conceptual shift from rigid, heuristic-based scaling to principled, utility-aware inference. By empowering models to anticipate their success and computational cost, our work is a key step toward more autonomous, reliable, and efficient LLMs. A limitation of our method is that we rely on LLMs achieving sufficient diversity of samples during inference; namely, if we double the number of initial samples, but the new samples are not sufficiently different, then our method and any similar test-time compute method like BoN is unable to achieve higher performance. We believe an important direction of future work is investigating how to improve diversity of samples during inference, potentially by using a mixture of prompts or even models. Overall, we believe ZIP-RC establishes a strong foundation for the next generation of introspective models and provides a timely, impactful contribution to adaptive test-time scaling.

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

## A APPENDIX

### A.1 SAMPLING AS DECISION-MAKING UNDER A META-MDP

We formalize the problem of test-time search as decision-making under a high-level MDP that we dub a *meta-MDP*. We describe the components of the meta-MDP in detail below:

**Meta-states.** At timestep $t$, the meta-state is a prefix tree (trie) $S_t$ rooted at the prompt $\mathbf{x}$. Formally, $S_t = (\mathcal{N}_t, \mathcal{E}_t, r)$ where $\mathcal{E}_t \subseteq \mathcal{N}_t \times \mathcal{V} \times \mathcal{N}_t$ is the set of directed edges $(s, a, s')$ labeled by tokens $a \in \mathcal{V}$, $r$ is the root corresponding to the prompt, and each node $s \in \mathcal{N}_t$ represents the prefix given by the concatenation of the root and the token labels along the path from $r$ to $s$. Each node also corresponds to a state in the base MDP as it is a sequence of tokens that the base policy has generated. A prefix is finished if its last edge is the special token <EOS>, corresponding to the terminal state in the base MDP. Initially, $S_0 = (\{r\}, \emptyset, r)$ where $r$ is the root containing the prompt. Conceptually, the meta-state therefore encodes all prefixes processed or generated by the policy. In practice, the trie only requires space on the order of the total tokens generated to represent all sequences, and their corresponding prefixes can be stored in the KV cache.

**Meta-actions.** At step $t$, the meta-action selects a finite multiset of prefixes (nodes) $A_t \subseteq \mathbb{M}(\mathcal{N}_t)$ to continue sampling from. Multiplicity encodes branching: if prefix $s$ appears $r$ times in $A_t$, then $s$ is sampled from $r$ times independently. This definition encodes any viable single-token sampling and any strategy one might want to perform. If $s \in A_{t-1}$ but none of its children appear in $A_t$, this is equivalent to pruning. If $s \in A_t$ already has children, this is equivalent to backtracking.

**Meta-transition function.** Given $(S_t, A_t)$, for each occurrence of $s \in A_t$ we sample $a \sim \pi(\cdot \mid s)$ and add the edge $(s, a, s')$ and its child $s'$ to the trie, yielding $S_{t+1}$. For notational simplicity, we treat $\pi$ as part of the transition dynamics of the meta-MDP so the transition function $P(S_{t+1} \mid S_t, A_t)$ implicitly includes sampling from $\pi$. The process terminates at horizon $T$, corresponding to the maximum allowed search steps.

**Meta-reward function.** At each step we incur a cost $C(S_t, A_t) = \beta\big(\alpha \,|\mathrm{supp}(A_t)| + (1 - \alpha)\big)$, where $\mathrm{supp}(A_t)$ is the set of distinct prefixes chosen in $A_t$ (one forward pass per unique prefix) and the second term accounts for step latency. The parameter $\alpha \in [0, 1]$ balances compute versus latency, and $\beta > 0$ sets the trade-off between reward and cost. At the terminal timestep $T$, one completed generation $s_T^*$ is selected, and the reward is the base MDP reward $R(s_T^*)$. Including this cost term is essential, since otherwise one could trivially maximize reward by always branching.

**Search strategies as meta-policies.** A strategy $\mu$ is a policy in this meta-MDP: at each timestep it maps the current prefix tree $S_t$ to a multiset $A_t = \mu(S_t)$ of nodes to expand, and at horizon $T$ selects a finished $f^*$. BoN corresponds to placing $N$ copies of the root in $A_0$ and thereafter always expanding every unfinished leaf until completion, finally selecting the highest-scoring candidate. Beam search with width $B$ instead enforces $|A_t| = B$ at all times: at most steps $A_t$ is just the $B$ current leaves, but at pruning intervals of length $k$ it ranks leaves by a score, discards the weakest $p$, and duplicates stronger ones so that the frontier is refilled back to $B$, thereby pruning and branching in a controlled manner before ultimately selecting the best finished prefix at $T$.

### A.2 COMPUTING AN OPTIMAL ZERO-OVERHEAD SEARCH STRATEGY

It is clear from our formalism what an optimal search strategy should be: at every timestep $t$, the optimal strategy $\mu^*$ should choose the meta-action that maximizes:

$$\mu^*(S_t) = \arg\max_{A_t} Q^{\mu^*}(S_t, A_t), \tag{9}$$

where we define a *meta* Q-function over the meta-MDP for a strategy $\mu$ as,

$$Q^\mu(S_t, A_t) = \mathbb{E}_\mu\left[ R(s_T^*) - \sum_{t'=t}^{T-1} C(S_{t'}, A_{t'}) \mid S_t, A_t \right]. \tag{10}$$

However, computing $Q^\mu$ for arbitrary strategies $\mu$ is often intractable, primarily because we cannot generate on-policy trajectories from $\mu$ without incurring too much computational overhead.

**The sampling utility.** To avoid having to generate rollouts, we consider a class of *predefined* strategies at any timestep $t$ as follows:

$$M_t = \{\mu : \mu(\cdot \mid S_{t'}) = f\left(\{\pi(\cdot \mid s_t)\}_{s_t \in \mathcal{N}_t}\right), \ \forall t' \geq t\}, \tag{11}$$

where $f$ denotes any function of the set of next-token distributions for every prefix in the meta-state. Concretely, $M_t$ consists of all strategies where future meta-actions are determined entirely from generation behavior at timestep $t$. This essentially means for $\mu \in M_t$, we can compute its value $Q^\mu$ without explicitly executing $\mu$ over future timesteps.

For any meta-state and meta-action at timestep $t$, we define the *sampling utility* to be the value of some strategy in the aforementioned class of strategies $M_t$. Because each strategy performs worse than optimal strategy $\mu^*$ due to the imposed constraint, we choose the best-performing strategy in $M_t$ to act as the tightest possible lower-bound

$$\mathcal{U}(S_t, A_t) = \max_{\mu \in M_t} Q^\mu(S_t, A_t) \leq Q^{\mu^*}(S_t, A_t). \tag{12}$$

We show later how this maximization over $M_t$, as well as computation of $Q^\mu$ for $\mu \in M_t$, can be done tractably using the quantities obtained via ZIP-RC, *without any additional forward passes, auxiliary models, or architectural modifications* beyond standard decoding.

Finally, ZIP-RC sampling is defined as the strategy that maximizes our proposed sampling utility:

$$\mu^{\text{ZIP-RC}}(S_t) = \arg\max_{A_t} \mathcal{U}(S_t, A_t). \tag{13}$$

Intuitively, we can derive the following property of our learned strategy:

**Theorem A.1.** *At every timestep $t$, our strategy $\mu^{\text{ZIP-RC}}$ performs better than any predefined strategy $\mu \in M_t$. Namely, for any meta-state $S_t$, we have*

$$Q^{\text{ZIP-RC}}(S_t, \mu^{\text{ZIP-RC}}(S_t)) \geq Q^\mu(S_t, \mu(S_t)), \ \forall \mu \in M_t. \tag{14}$$

*Proof.* We can prove this via induction on $t$. Naively, this holds for terminal timestep $t = T$. For any $\mu \in M_t$, we let $A_t^{\text{ZIP-RC}} = \mu^{\text{ZIP-RC}}(S_t)$ and $A_t^\mu = \mu(S_t)$. Then, we have

$$Q^{\text{ZIP-RC}}(S_t, A_t^{\text{ZIP-RC}}) = Q^{\text{ZIP-RC}}(S_{t+1}, A_{t+1}^{\text{ZIP-RC}}) - C(S_t, A_t^{\text{ZIP-RC}}). \tag{15}$$

$\square$

Therefore, ZIP-RC sampling is a powerful test-time search strategy that explicitly optimizes for reward and generation cost.

**Approximating the sampling utility.** To approximate the sampling utility, we aim to answer two questions: (1) for every meta-state $S_t$ and action $A_t$, how do we search for a strategy $\mu \in M_t$ that achieves a high value $Q^\mu(S_t, A_t)$, and (2) how do we compute $Q^\mu(S_t, A_t)$ tractably using only predictions by ZIP-RC.

First, let us consider the naive strategy $\mu^{\text{Rollouts}}$ of always selecting the unfinished leaf-node descendants of the prefixes in the current action $A_t$, or in other words, obtaining *rollouts* or generations using $\pi$ starting from each selected prefix. At the end, $\mu^{\text{Rollouts}}$ selects the generation with the highest value $V^\pi(s_T)$, similar to BoN. Its meta-MDP state-action value is exactly given by:

$$Q^{\text{Rollouts}}(S_t, A_t) = \mathbb{E}_{\mu^{\text{Rollouts}}}\left[R(s_T^*) - \sum_{t'=t}^{T-1} C(S_{t'}, A_{t'}) \mid S_t, A_t\right] \tag{16}$$

$$= \mathbb{E}\left[\max_{s \in A_t} V_T^\pi(s) - \beta\left(\alpha \sum_{s \in A_t} L_T^\pi(s) + (1-\alpha)\max_{s \in A_t} L_T^\pi(s)\right)\right]$$

$$= \mathbb{E}\left[\max_{s \in A_t} V_T^\pi(s)\right] - \beta\left(\alpha \sum_{s \in A_t} \mathbb{E}[L_T^\pi(s)] + (1-\alpha)\mathbb{E}\left[\max_{s \in A_t} L_T^\pi(s)\right]\right). \tag{17}$$

We can observe that the expression $Q^{\text{Rollouts}}$ contains several interpretable quantities. Namely, the expected maximum value quantifies the marginal benefit of branching or pruning; the incremental gain from increasing $N$ is large when the value distribution has high variance, and conversely, the marginal loss of pruning is small under low variance. Furthermore, the expected maximum remaining tokens and the expected total tokens capture the marginal cost of branching; increasing $N$ always increases the expected total remaining tokens and the maximum remaining length, which drives up latency, and pruning will always reduce the cost.

While $\mu^{\text{Rollouts}}$ has several nice properties, the strategy itself is naive as it assigns maximum cost for every new sample and does not consider that those samples can be pruned in the future. This is exacerbated further by the empirical correlation between the length of reasoning traces and the likelihood that they are incorrect. Being able to "bet" on an early finishing sample that has high reward is crucial. To remedy this problem, we introduce an additional parameter into the meta-value that enables the $\mu^{\text{Rollouts}}$ strategy to prune each sample in the future at a predefined horizon.

Formally, let $Q^{\text{Rollouts}}(S_t, A_t; \mathcal{H}_t)$ be the value of executing $\mu^{\text{Rollouts}}$, with the additional capability that each active prefix $s \in A_t$ will stop generating upon reaching length $h_s \in \mathcal{H}_t$, where $\mathcal{H}_t$ is a set of lengths of size $|A_t|$. We can now define an improved lower bound by maximizing over $\mathcal{H}_t$:

$$Q^{\text{Rollouts}}(S_t, A_t; \mathcal{H}_t^*) = \max_{\mathcal{H}_t} Q^{\text{Rollouts}}(S_t, A_t; \mathcal{H}_t) \geq Q^{\text{Rollouts}}(S_t, A_t), \tag{18}$$

where it is easy to see that the our new meta-value is monotonically better than the value of the naive $\mu^{\text{Rollouts}}$ strategy without pruning capabilities. We use this to approximate the sampling utility:

$$\mathcal{U}(S_t, A_t) = Q^{\text{Rollouts}}(S_t, A_t; \mathcal{H}_t^*). \tag{19}$$

**Tractable computation of expectations.** Next, we show how our sampling utility in Equation 19 can be computed tractably using only predictions by ZIP-RC. Predicting $Q^{\text{Rollouts}}$ is straightforward since we can estimate distributions over the value $q_\theta^V(\cdot|s_t)$ and remaining-tokens $q_\theta^L(\cdot|s_t)$ conditioned on each prefix $s_t$ using our previously proposed ZIP-RC. Hence, we can estimate by computing:

$$\mathbb{E}\left[\max_{s \in A_t} V_T^\pi(s)\right] \approx \sum_{b=1}^{B_V} \frac{v_b + v_{b+1}}{2} \left(F_\theta^{V,\max}(b \mid A_t) - F_\theta^{V,\max}(b - 1 \mid A_t)\right), \tag{20}$$

$$\mathbb{E}[L_T^\pi(s)] \approx \sum_{\ell=1}^{B_T} \frac{t_\ell + t_{\ell+1}}{2} q_\theta^L(\ell \mid s), \tag{21}$$

$$\mathbb{E}\left[\max_{s \in A_t} L_T^\pi(s)\right] \approx \sum_{\ell=1}^{B_T} \frac{t_\ell + t_{\ell+1}}{2} \left(F_\theta^{L,\max}(\ell \mid A_t) - F_\theta^{L,\max}(\ell - 1|A_t)\right), \tag{22}$$

where

$$F_\theta^{V,\max}(b \mid A_t) = \prod_{s \in A_t} F_\theta^V(b \mid s), \qquad F_\theta^{L,\max}(\ell \mid A_t) = \prod_{s \in A_t} F_\theta^L(\ell \mid s), \tag{23}$$

$$F_\theta^V(b \mid s) = \sum_{j \leq b} q_\theta^V(j \mid s), \qquad F_\theta^L(\ell \mid s) = \sum_{j \leq \ell} q_\theta^L(j \mid s). \tag{24}$$

Now, to incorporate the predefined horizons $\mathcal{H}_t$ over all prefixes $A_t$, we modify the joint distribution $p_\theta(b, \ell|s)$ to a *capped* joint $p_\theta(b, \ell|s; h_s)$ that collapses probability mass beyond the cap $h_s$ into a designated "clipped" state $(b_0, h_s)$:

$$p_\theta(b, \ell \mid s; h_s) = \begin{cases} p_\theta(b, \ell \mid s), & \ell \leq h_s, \\ \mathbf{1}\{b = b_0, \ell = h_s\} \displaystyle\sum_{b'} \sum_{\ell' > h_s} p_\theta(b', \ell' \mid s), & \ell > h_s. \end{cases} \tag{25}$$

This construction ensures that all probability mass corresponding to continuations exceeding the allowed horizon is reassigned to a truncated state at $\ell = h_s$, while the value component is collapsed to the designated base bin $b_0$ to reflect the forfeited reward from pruning.

From the capped joints, we can recover the corresponding marginal distributions:

$$q_\theta^V(b \mid s; h_s) = \sum_{\ell=1}^{B_T} p_\theta(b, \ell \mid s; h_s), \qquad q_\theta^L(\ell \mid s; h_s) = \sum_{b=1}^{B_V} p_\theta(b, \ell \mid s; h_s). \qquad (26)$$

These capped marginals directly encode the expected effect of planned pruning on both value and remaining length for each prefix. Notice that this is only possible by modeling the joint distribution as ZIP-RC is defined and is not possible with only the two marginal distributions. Thus, we demonstrate that we are able to compute our sampling utility in Equation 19 using only our zero-overhead predictions from ZIP-RC.

**Summary.** ZIP-RC sampling defines a meta-policy that, at each meta-state $S_t$, selects the meta-action $A_t$—a multiset of prefixes to expand for one decoding step—that maximizes the *sampling utility* in eq. (19). This utility is the state–action value of the best policy in the predefined strategy class $M_t$, whose future behavior is fixed. Because ZIP-RC sampling re-optimizes $A_t$ at every timestep, it adapts online to the stochastic evolution of the prefix tree: if current trajectories are projected to be costly or low-value, it can immediately redirect computation elsewhere. As formalized in eq. (14), this dynamic strategy is guaranteed to perform at least as well as any predefined policy in $M_t$.

To approximate the sampling utility tractably, we use the value of the best rollouts-with-pruning strategy $Q^{\text{Rollouts}}(S_t, A_t; \mathcal{H}_t^*)$, in which each prefix may continue only up to an optimized (and possibly distinct) horizon. This value can be computed in closed form using ZIP-RC's joint reward–cost predictions. Concretely, for every prefix $s \in A_t$, we: (i) obtain its predicted joint distribution $p_\theta(b, \ell \mid s)$; (ii) apply the prefix-specific horizon using the capped construction in eq. (26); and (iii) compute the expectations of the required order statistics using eq. (22). This yields a fully tractable, zero-overhead estimate of the sampling utility for any candidate meta-action.

### A.3 ZIP-RC IMPLEMENTATION DETAILS

**Remaining-token discretization.** For the joint ZIP-RC head, we discretize the remaining-length variable $L_T^\pi(s_t)$ using logarithmic bins. Let $\{t_\ell\}_{\ell=1}^{B_T+1}$ denote the bin boundaries, and define bin $\ell$ as $[t_\ell, t_{\ell+1})$ with representative value $(t_\ell + t_{\ell+1})/2$. To obtain fine resolution for short continuations while keeping $B_T$ small, we collapse all very short lengths into a single initial "startup" bin and set the remaining boundaries to grow as powers of two. This construction preserves precision where it matters while limiting the number of reserved tokens required for ZIP-RC.

**KL weight.** In practice, the KL term is a very small component of the total loss because the policy remains close to the original policy as seen in fig. 4. Accordingly, we use a relatively large coefficient $\alpha_{\text{KL}}$, typically in the range 10–100.

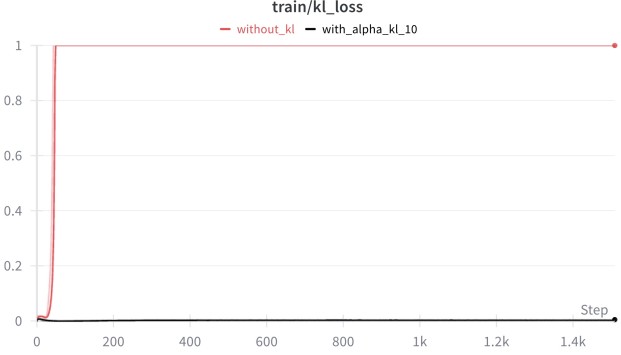

Figure 4: KL divergence from the original policy during training of ZIP-RC with and without the KL term. Using $\alpha_{\text{KL}} = 10$ keeps the KL nearly zero throughout training, stabilizing around 0.005. Without the KL term, the policy eventually changes, emphasizing the importance and effectiveness of this component of the ZIP objective. We used the same training data as in our main experiments.

**Temporal smoothing.** Token-level ZIP-RC predictions can be noisy. At inference time we optionally smooth the joint by averaging the last $W$ predictions along the current trajectory: for a prefix $s_t$ we set

$$\bar{p}_\theta(b, \ell \mid s_t) = \frac{1}{W} \sum_{w=0}^{W-1} p_\theta(b, \ell \mid s_{t-w}), \tag{27}$$

where the sum runs over the previous non-terminal prefixes on the same path. We use $\bar{p}_\theta$ in place of $p_\theta$ when computing the ZIP-RC marginals and the sampling utility.

## A.4 ZIP-RC SAMPLING IMPLEMENTATION DETAILS

**Normalization of cost term $\beta$.** Because rollout lengths can differ by orders of magnitude across prompts, we rescale the cost coefficient $\beta$ by a per-prompt estimate of the typical total token count. At decision step $t$, we use the normalized coefficient

$$\tilde{\beta}_t = \frac{\beta}{\bar{B}_t}, \qquad \text{where} \qquad \bar{B}_t = \frac{1}{|A_t|} \sum_{s \in A_t} \left( |s| + \mathbb{E}[L_T^\pi(s)] \right). \tag{28}$$

Here $|s|$ is the current length of prefix $s$, and $\mathbb{E}[L_T^\pi(s)]$ is its predicted remaining tokens from ZIP-RC. This keeps the reward–cost tradeoff stable across prompts with very short or very long generations.

**Practical reduction of the search space.** The full meta-action space over multisets of nodes in the prefix tree is extremely large, and jointly optimizing per-prefix horizons is combinatorial. In our implementation we therefore operate within a structured subclass of meta-actions and horizons. First, at timestep $t$ we restrict candidate meta-actions to multisets over the root and the unfinished leaves of $S_t$ with multiplicity only allowed at the root, and further downselect to a small set of prefixes prioritized by higher predicted value and lower predicted remaining length under ZIP-RC. Second, when computing the sampling utility we use a single shared horizon $h_t$ for all active prefixes, rather than independent horizons per prefix, reducing the search over pruning schedules to a one-dimensional search over $h_t$. Finally, instead of recomputing $\mu^{\text{ZIP-RC}}(S_t)$ at every token, we update the meta-action only at fixed intervals and simply continue all currently active prefixes in between. These design choices substantially reduce the search space while preserving an expressive and adaptive family of strategies. This represents just one concrete instantiation of our framework; many other reductions are possible.

## A.5 ZIP-RC-LITE

We add an ablation we refer to as ZIP-RC-Lite where we use the ZIP-RC objective described in eq. (3) with the KL term removed, keeping only the output head of the language model trainable while freezing the rest of the model. As shown in fig. 5 and table 3, we find that ZIP-RC-Lite is able to non-trivially predict the joint distribution but unsurprisingly struggles to do so as accurately as ZIP-RC. Its predictions are poorly calibrated and may not be suitable for interpretability of the generation process and output. Despite this, we find that ZIP-RC-Lite search provides substantial gains with respect to baselines in the latency-bound setting with $\alpha = 0.1$, suggesting that predicting the expected reward and remaining length to any degree is useful for pruning overly long trajectories as seen in the first row of fig. 6. However, we do find that ZIP-RC-Lite substantially over-allocates compute in the compute-bound setting with $\alpha = 1.0$ due to overestimating the variance of the expected reward, resulting in poor efficiency as seen in the second row of fig. 6. Overall, while ZIP-RC is clearly more calibrated and accurate, ZIP-RC-Lite is a compelling alternative if one does not want to keep the whole model trainable.

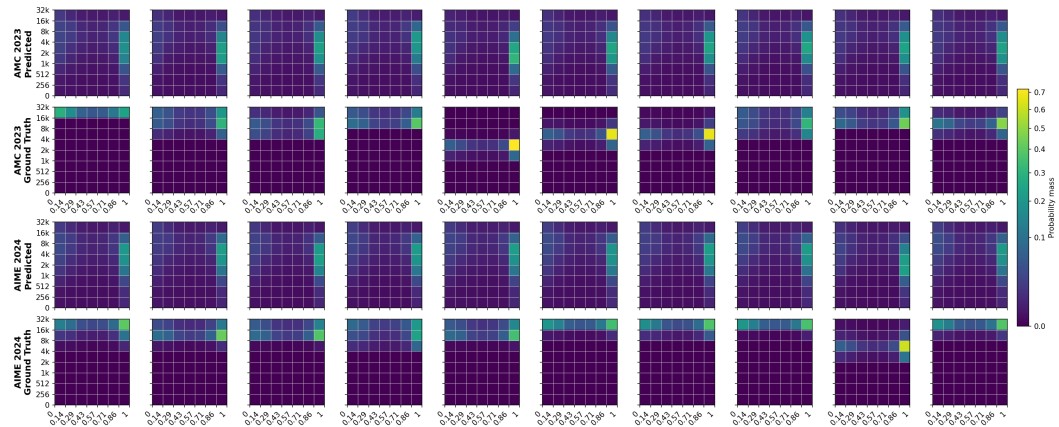

Figure 5: Similar to the demonstration of ZIP-RC's joint distribution prediction in fig. 2, we visualize the joint distribution predictions from ZIP-RC-Lite and compare them with ground truth estimates. While the predictions correlate with the ground truth, ZIP-RC-Lite tends to produce more similar-looking distributions across prompts and overestimates variance compared to ZIP-RC.

| | Beginning (Reward+Cost) | End (Reward) | | |
|---|---|---|---|---|
| Method | Total Variation | F1 Score | Accuracy | Recall (Incorrect) |
| ZIP-RC | 0.46 | 0.91 | 0.88 | 0.82 |
| ZIP-RC-Lite | 0.63 | 0.82 | 0.71 | 0.12 |

Table 3: Similar to the evaluation of ZIP-RC in table 1, we show the prediction accuracy of ZIP-RC-Lite at the beginning and end of generation. The results indicate that ZIP-RC-Lite predicts the joint distribution non-trivially, but less accurately than ZIP-RC.

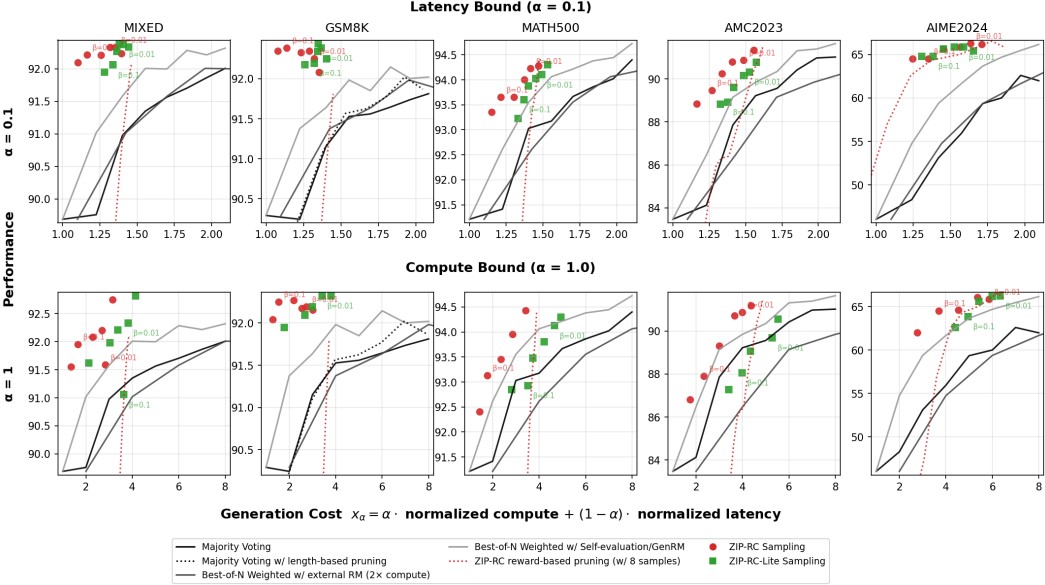

Figure 6: Similar to the demonstration of ZIP-RC sampling in fig. 3, we present results with ZIP-RC-Lite search in green, indicating it is able to provide significant gains, albeit lower than ZIP-RC sampling, especially in the compute-bound setting. This suggests that predicting the joint reward-cost distribution to any non-trivial degree is helpful in allocating test-time compute more optimally. However, the results in the compute-bound setting indicate that ZIP-RC-Lite's overestimation of variation in the expected reward distribution results in over-allocation of compute.

