# OpenReview forum: "Zero-Overhead Introspection for Adaptive Test-Time Compute"
_ICLR.cc/2026/Conference — ICLR 2026 Poster_

### Official Review · Reviewer_5e4c · 2025-11-01

**Soundness:** 2
**Presentation:** 2
**Contribution:** 3
**Rating:** 6
**Confidence:** 3

**Summary:**

This work is in the context of autoregressive Language Models with a finite vocabulary (i.e. popular transformer-based LLMs).
The paper aims to improve the model's introspective ability (e.g., predicting at each token, prediction final
reward and remaining length) without increasing the test time inference cost.
To enable this, the paper proposes to use a reserved or unused set of logits and thus claims to require no additional compute at inference time.
The paper examines this within the framework of test-time scaling, i.e., using test-time compute for improved performance.

**Strengths:**

1. The proposed approach of using unused logits in combination with the proposed optimization of a joint distribution different from the next token prediction loss (but over an estimated value of the correctedness and remaining tokens) seems very interesting and innovative to me.
2. The results of the proposed approach in comparison to the presented baselines show clear benefits on the Pareto curve.

**Weaknesses:**

## Major weaknesses
1. The paper's clarity in the main approach section is limited. Sections 4.1 and 4.2 are difficult to parse and would benefit from more careful dissemination, including model/approach figures.
2. The paper claims **zero** inference time overhead. However, computations described in Section 4.2, paragraph "Sampling utility." suggest that additional computations and parameters (alpha, beta, ...) are required (even though they may be small compared to the overall computation and parameter requirements of the LLM). It would be important to clarify this.
4. The paper's approach heavily relies on a reserved set of logits but does not describe where they come from. Would we save compute if we are not using these "reserved" logits? This also relates to the previous point: Do these reserved logits basically represent additional computation needed for this approach
3. I think the paper is missing an important baseline, which does not change the underlying LLM (i.e. one does not need the KL term), but just optimizes the same reward and cost (ZIP-RC) using a small MLP (one? or two layers), which has access to the same logits used for the proposed approach.


## Minor weaknesses
1. all formulas should be numbered so one can refer to them

**Questions:**

Please comment on / address the weaknesses above.

Additionally:
1. line 267: "ZIP-RC modifies only the sampling policy at inference-time." This seems misleading as the model does train and thus changes the parameters of the original models.
2. How many reserved logits are used in practice?
3. What is the initial number of samples? And how does this compare to baselines?
3. What other use case do the authors see for this framework (reserved logits for a different reward) besides the presented application of test-time scaling?

---

> ### Author Response · Authors · 2025-11-26
> **Response to Reviewer 5e4c**
>
> Thank you for your review. We appreciate that you found our approach "innovative" and recognized the clear benefits on the Pareto curve. **To directly address your concerns regarding clarity and baselines, we have substantially restructured the paper, added a comprehensive overview figure, and implemented the exact ablation you suggested.** We added **Figure 1** to visualize the end-to-end mechanism, clarified the definition of "zero-overhead" to distinguish model vs. controller compute, and added **ZIP-RC-Lite** (a frozen-backbone baseline) to isolate the effect of the method without modifying the underlying LLM. We believe these changes resolve the ambiguity in the original submission.
>
> **Q1: The method sections (4.1, 4.2) were difficult to parse and lacked visual aids.**
> **A1:** We agree and have overhauled the presentation. Please see **Overall Response Points 1 & 3**. We split the original text into **Section 4** (The ZIP mechanism and training) and **Section 5** (A high-level overview of ZIP-RC Sampling). To improve readability and manage length, the formal definitions of the Meta-MDP and the derivation of the sampling utility have been moved to **Appendices A.1 and A.2**. Crucially, we added **Figure 1** in the main text, which provides the diagram you requested: it illustrates the logit repurposing, the grid mapping, and the resulting flow from prediction to adaptive sampling.
>
> **Q2: The claim of "zero inference time overhead" seems contradictory if the sampling utility requires computation (e.g., optimizing with $\alpha, \beta$).**
> **A2:** We have clarified this distinction in the revision. Please see **Overall Response Point 2**. We define "zero-overhead" specifically regarding *model compute* (GPU forward passes and parameters). The sampling utility calculation (derived in **Appendix A.2**) is a lightweight CPU-side operation over scalar values (predicted rewards and lengths) performed by the controller. Compared to the massive cost of an LLM forward pass, this control logic is negligible, but we now explicitly distinguish between the "free" model predictions and the lightweight controller logic.
>
> **Q3: Where do the reserved logits come from, and does generating them require extra compute?**
> **A3:** We have clarified this in **Section 4**. As noted in **Overall Response Point 2**, standard LLM decoding computes the projection onto the *entire* vocabulary at every step. ZIP-RC simply designates a small subset of these already-computed logits (the reserved set $\mathcal{R}$) as the prediction head. Therefore, extracting these signals requires zero additional FLOPs compared to standard generation.
>
> **Q4: The paper is missing a baseline that uses the same logits/objective but freezes the LLM (no KL term), effectively using a small probe.**
> **A4:** We have implemented this exact baseline. Please see **Overall Response Point 4**. We added **ZIP-RC-Lite** (**Appendix A.5**), which freezes the transformer backbone and trains only the output head (reserved logits) without the KL term. As shown in the new **Figure 5** and **Figure 6**, this variant works but is less calibrated than full ZIP-RC. This confirms your intuition that the mechanism is valid even without modifying the base model, though fine-tuning yields better calibration.
>
> **Q5: The text "ZIP-RC modifies only the sampling policy" is misleading since the model is trained.**
> **A5:** You are correct. We have revised **Section 6.1** to state clearly that ZIP-RC involves a one-time training phase (modifying model weights), and that *at inference time* it modifies the sampling policy using the learned signals.
>
> **Q6: Please number all formulas.**
> **A6:** We have done so. All key equations (Best-of-N, ZIP loss, Joint Distribution, Sampling Utility) are now numbered for easy reference.
>
> **Q7: Can you provide implementation details regarding the number of reserved logits and samples?**
> **A7:** We have added these details to the text. As noted in **Overall Response Points 1 & 3**, we use:
> * **Reserved Logits:** 64 tokens (an 8x8 grid for Reward $\times$ Remaining Length).
> * **Samples:** A maximum pool of 8 samples per prompt (adaptive), which allows fair comparison to baselines that use fixed budgets of 2–8 samples.
>
> **Q8: What other use cases exist for this framework?**
> **A8:** As mentioned in **Overall Response Point 1**, ZIP is a general interface for introspection. Beyond test-time scaling, it can be used to predict safety scores (toxicity), hallucination probabilities, or retrieval relevance at every token step without slowing down generation, serving as a lightweight "heads-up display" for the model's internal state.

---

### Official Review · Reviewer_VsZ8 · 2025-11-01

**Soundness:** 3
**Presentation:** 3
**Contribution:** 3
**Rating:** 6
**Confidence:** 4

**Summary:**

This paper tackles an important limitation of current test-time scaling methods for large language models: their inability to adaptively allocate compute based on problem difficulty. The authors propose ZIP-RC, a method that equips models with real-time predictions of both final reward and remaining generation cost without any inference overhead. The key innovation is repurposing reserved vocabulary logits to output a joint distribution over reward and cost at every decoding step. This enables optimizing a sampling utility that explicitly trades off accuracy against compute and latency through meta-actions like dynamic pruning and adaptive sample allocation. On mathematical reasoning benchmarks, ZIP-RC achieves up to 12% accuracy improvements over majority voting while using equal or lower cost, and traces smooth Pareto frontiers across quality, compute, and latency dimensions.

**Strengths:**

1. The idea of using reserved vocabulary positions to produce auxiliary predictions with truly zero overhead is novel and elegant. Rather than requiring separate forward passes or additional models like most verifier approaches, ZIP-RC extracts rich signals from logits that would otherwise go unused. The joint modeling of reward and cost (rather than just scalar confidence) is a key insight that enables principled decision-making about the reward-cost tradeoff.

2. The authors clearly articulate why existing Best-of-N methods are inefficient—they generate all N samples to completion regardless of promise. The mathematical formulation of the sampling utility in Equation 1 provides an interpretable framework for balancing competing objectives through the α and β hyperparameters.

3. The paper is generally well-written with clear motivation and good use of figures to illustrate the method's behavior. The formalization as a token-level MDP and the discrete order statistics for computing expectations are technically sound.

**Weaknesses:**

1. The evaluation focuses exclusively on mathematical reasoning tasks. It's unclear whether ZIP-RC's benefits extend to other domains like creative writing, coding, or open-ended question answering where the reward structure and token length distributions may be very different. Mathematical problems have clear correctness labels and relatively predictable structure, which may make reward/cost prediction easier than in other domains.

2. The authors acknowledge that their method relies on having sufficiently diverse samples from the initial pool, but they don't actually address this limitation. If doubling the number of samples doesn't produce qualitatively different reasoning paths, ZIP-RC cannot improve performance. This is a fundamental constraint that limits when the method can be applied effectively. Some analysis of sample diversity in their experiments would strengthen the work.

3. The paper doesn't clearly specify how many vocabulary positions are reserved, how this impacts the model's generative capabilities, or what the training cost is. The KL penalty term αKL is mentioned but its practical effect isn't analyzed. More importantly, ZIP-RC requires generating ~100k labeled rollouts and training the predictor—this upfront cost should be quantified and compared to the inference savings.

**Questions:**

1. How many positions are reserved for the reward-cost predictor? How does this choice affect both the predictor's expressiveness and the base model's generation quality? Have you experimented with different numbers?

2. Can you provide any evidence that ZIP-RC works on non-mathematical tasks? Even preliminary results on a single coding or open-ended QA benchmark would significantly strengthen the generalization claims.

3. What is the wall-clock time and compute cost to train ZIP-RC compared to training the baseline models? How much data is needed for the predictor to become reliable?

---

> ### Author Response · Authors · 2025-11-26
> **Response to Reviewer VsZ8**
>
> Thank you for your review. We appreciate that you found the zero-overhead mechanism "novel and elegant" and the sampling utility framework "interpretable" and "technically sound." **To address your concerns regarding generalization and implementation transparency, we have added explicit analyses of the training components (KL term, reserved tokens)** and are extending our evaluation to a non-mathematical domain. We believe these additions solidify the empirical grounds of the method.
>
> **Q1: Can you demonstrate that ZIP-RC generalizes to domains beyond mathematics, such as coding or open-ended QA?**
> **A1:** We agree that an additional domain will strengthen the evidence. We are actively extending our analysis to a science domain using **GPQA-D** (constructed from the OpenScience dataset). This experiment utilizes the exact same ZIP-RC training pipeline as the math benchmarks. We will include the resulting calibration plots and Pareto frontiers in the appendix to demonstrate that ZIP-RC, which abstracts tasks into reward and length predictions, is not specific to mathematical reasoning.
>
> **Q2: How does the reliance on sample diversity limit the method, and can you quantify this?**
> **A2:** We agree that diversity is the fundamental bottleneck for all sampling-based methods (including BoN and Self-Consistency). We now explicitly discuss this in the **Conclusion** and **Section 6.4**. Empirically, this limitation is visible in **Figure 3** (e.g., on the "Mixed" benchmark), where performance saturates once the model exhausts the diversity of its high-quality reasoning paths. We clarify that ZIP-RC is designed to optimize the *allocation* of these paths, and we outline in the discussion how it can be combined with diversity-enhancing techniques (e.g., prompt ensembling) in future work.
>
> **Q3: Please clarify implementation details regarding the number of reserved tokens, the practical impact of the KL term, and the training cost relative to inference savings.**
> **A3:** We have added these specific details to the revision:
> * **Reserved Tokens:** We use an **8x8 grid (64 tokens)**. This design choice and the discretization scheme are now detailed in **Figure 1** and **Appendix A.3**.
> * **KL Term:** Please see **Overall Response Point 5**. We added **Figure 4 (Appendix A.3)**, which plots the KL divergence during training. It shows that with our regularization ($\alpha_{KL}=10$), divergence stabilizes near zero (~0.005), effectively preventing the base policy from drifting.
> * **Training Cost:** We clarify in **Section 6.1** that ZIP-RC shares the same training data requirement (~100k labeled rollouts) as the standard Verifier/Reward Model baselines. The training cost is a one-time upfront investment comparable to training a standard verifier, but it enables the Pareto-dominant inference efficiency shown in **Table 2** and **Figure 3**.

---

### Official Review · Reviewer_5iv6 · 2025-11-01

**Soundness:** 3
**Presentation:** 2
**Contribution:** 4
**Rating:** 8
**Confidence:** 3

**Summary:**

This paper proposes ZIP-RC, a zero-overhead inference-time mechanism that equips large language models with real-time predictions of task reward and remaining generation length. The key idea is to utilize a set of reserved vocabulary tokens as auxiliary prediction channels, allowing the model to output both next-token probabilities and reward/length estimates within the same forward pass. To prevent distribution shift, the method introduces a KL-regularization term that constrains the model’s behavior relative to a frozen reference policy.
On top of these predictions, the paper introduces an adaptive sampling and pruning strategy that dynamically reallocates compute across candidate trajectories, enabling the model to prune low-quality or unnecessarily long samples mid-generation. This yields controllable trade-offs between output quality, compute cost, and latency, improving inference-time efficiency.
Experiments on mathematical reasoning benchmarks (GSM8K, AIME24, AMC23, MATH-500) and small-to-medium LLMs (≤1.7B) show that ZIP-RC consistently outperforms majority voting and best-of-N baselines under equal or lower generation cost, demonstrating more favorable quality-compute Pareto frontiers.

**Strengths:**

1.The paper proposes the ZIP-RC mechanism, which leverages reserved tokens in the vocabulary to enable real-time prediction of reward and remaining tokens without adding extra forward passes. This idea is highly innovative. Combined with adaptive pruning, the approach significantly improves inference efficiency and is of substantial practical importance.

2.Experimental results show that ZIP-RC clearly outperforms the baseline methods without such modification, demonstrating the strong potential of the proposed approach.

**Weaknesses:**

1.Although reserved tokens are used to avoid additional computation overhead, this strategy may still introduce distribution shift to some extent. The paper would benefit from additional comparison or analysis on the extent of distribution shift before and after modifying the loss function.

2.The experiments are primarily conducted on relatively small models (mostly under 2B). It would strengthen the work to extend evaluation to larger-scale models to verify scalability. Additionally, the experiments currently focus mainly on mathematical reasoning tasks; further results on other domains (e.g., instruction following, coding, and open-domain dialogue) are needed to validate generalization ability.

**Questions:**

Address the shortcomings identified in the Weaknesses section, and include the required supplementary experiments to adequately resolve these concerns and strengthen the study.

---

> ### Author Response · Authors · 2025-11-26
> **Response to Reviewer 5iv6**
>
> Thank you for your strong support and for recognizing ZIP-RC as "highly innovative" and of "substantial practical importance." We are glad you appreciated the zero-overhead mechanism and the efficiency gains. **To directly address your suggestions for strengthening the paper, we have added the specific analyses and experiments you requested** and are currently extending the evaluation to the domains and scales you suggested. We believe these additions fully resolve the weaknesses you identified.
>
> **Q1: Does the use of reserved tokens introduce distribution shift, and can you analyze the extent of this shift before and after modifying the loss?**
> **A1:** We have added the requested analysis. Please see **Overall Response Point 5**, where we describe our new monitoring of KL divergence between the ZIP-RC policy and the original frozen policy. As shown in the new **Figure 4 (Appendix A.3)**, our KL regularization term effectively keeps the divergence near zero (~0.005), preventing the distribution shift that occurs without it. Additionally, as detailed in **Overall Response Point 4**, we added **ZIP-RC-Lite** (frozen backbone) in **Appendix A.5**, which isolates the effect of the reserved logits from the backbone update, confirming that the mechanism works even with zero backbone shift (albeit with lower expressivity).
>
> **Q2: Can you extend the evaluation to larger-scale models to verify scalability?**
> **A2:** We agree that scalability is key. Our current results already show consistent gains across a 5x scale range (350M to 1.7B). To further confirm this trend, we are currently training ZIP-RC on **Qwen3-4B** and will report results on the AIME 2024 and AMC 2023 benchmarks in the final version of the paper.
>
> **Q3: Can you demonstrate generalization to domains beyond mathematical reasoning (e.g., science, coding)?**
> **A3:** We agree that this will strengthen the evidence and are acting on this suggestion. We are adding an experiment on a non-mathematical domain, specifically scientific QA, using **GPQA** constructed from the OpenScience corpus. This experiment reuses the exact same ZIP-RC training pipeline, demonstrating that our zero-overhead introspection mechanism generalizes to domains with different rewards.

---

### Official Review · Reviewer_hUjT · 2025-11-04

**Soundness:** 2
**Presentation:** 1
**Contribution:** 2
**Rating:** 2
**Confidence:** 4

**Summary:**

This paper proposes ZIP-RC, an adaptive inference method that equips model with zero-overhead inference-time predictions of reward and cost. ZIP-RC makes use of *reserved logits* in the forward pass to model a joint distribution over the reward as well as the remaining sequence. The authors show that ZIP-RC improves accuracy by up to 12% over majority voting at comparable computational cost.

**Strengths:**

- The promise of the paper, "Zero-overhead inference-time" control is a very promising and timely research direction

- The authors show that their approach is decently accurate at predicting the rewards during generation.

- The proposed approach achieves improvements, often significant, on a suite of reasoning benchmarks at the same cost as the baselines.

**Weaknesses:**

- I found the paper extremely difficult to read, and beyond the promise of a "zero-overhead inference-time prediction of reward" found it very hard to glean much if any insight on the core contributions of the paper beyond the fact it makes use of extra logits at every step.

- The related works section is really lacking giving how active an area of research inference-time control of LMs is.

- line 85, broken figure reference.

- Paragraph 074-085 of the introduction misses the mark when it comes to introducing the approach, in my opinion. Specifically, starting line 080, the paragraph makes mention of sampling utility and coefficients $\alpha$ and $\beta$ that supposedly balance reward and cost. This level of detail at this point of the paper left me more confused than informed, with no big picture idea of what the core contribution is.

- I am unsure if the subsection on "Generation as a token-level MDP" offers much value. In fact, it seems that variables defined in that section were used differently in later section e.g. $P$.

- Lines 170-171: "ZIP interprets the slice on $\mathcal{R}$". I believe the previous sentence should be "slice on $\mathcal{V}" instead?

- It took me quite some time to realize that the approach is *not* inference time, but instead relies on some form of training. In my opinion, the authors should make sure this is clear very early on in the paper as well as potentially clarify the title which to me is currently a bit misleading.

- The presented approach fails to compare against many of the latest inference-time approaches in their experiments

**Questions:**

- Could you please give me a breakdown of your technical approach? What exactly does your approach do at training time? How do you modify autoregressive decoding to model the joint distribution of sequence continuations and their corresponding rewards? A diagram would've been extremely helpful.

---

> ### Author Response · Authors · 2025-11-26
> **Response to Reviewer hUjT**
>
> Thank you for your review. We appreciate that you recognized the promise of zero-overhead control and our strong empirical results. We agree with your assessment that the original presentation made it difficult to identify the core contributions and technical flow. **To directly address this, we have extensively rewritten the paper to prioritize clarity.** We added a comprehensive **overview Figure 1** and **real examples in Figure 2** that illustrate the mechanism end-to-end, restructured the Methods section to clearly separate the training mechanism (ZIP) from the inference-time controller (ZIP-RC Sampling), and explicitly clarified the "zero-overhead" definition early in the Introduction. We believe these changes, along with the corrected notation and expanded baselines, resolve the readability issues and clearly present the technical contributions you found promising. We hope these revisions sufficiently address your concerns to warrant an improved score. Please let us know if you have any additional questions.
>
> **Q1: Can you provide a technical breakdown of the training vs. inference process and a diagram?**
> **A1:** We have added a new logical flow and visual aids to address this. Please see **Overall Response Point 1** regarding the new **Figure 1 and 2**, which provides the diagram you requested and real example predictions compared to ground truth. Additionally, as detailed in **Overall Response Point 3**, we reorganized the text so that **Section 4** now exclusively covers the training phase (repurposing reserved logits to learn the joint distribution). **Section 5** covers the inference phase (using those predictions to optimize the sampling utility), while the full formal derivations have been moved to **Appendices A.1 and A.2** to streamline the main text.
>
> **Q2: The paper was difficult to read, making it hard to find core contributions beyond the use of extra logits.**
> **A2:** We believe the restructuring described in **Overall Response Point 3** and the new visual overview in **Overall Response Point 1** resolve this. The revised Introduction now explicitly enumerates the three core contributions: (1) the zero-overhead joint prediction mechanism, (2) the sampling utility formulation for adaptive control, and (3) the resulting Pareto improvements.
>
> **Q3: The title and text were misleading regarding "inference-time," as the method requires training.**
> **A3:** We agree that this distinction must be precise. Please see **Overall Response Point 2**. We now explicitly state in the Abstract, Introduction, and Section 4 that ZIP-RC requires a one-time training phase (amortized over deployment), similar to training a verifier. We clarify that "zero-overhead" refers strictly to the inference deployment phase (no extra models or forward passes).
>
> **Q4: The Introduction introduced complex concepts (sampling utility, coefficients) too early, disrupting the flow.**
> **A4:** We have rewritten the Introduction to focus solely on high-level intuition. As noted in **Overall Response Point 3**, we removed the mathematical notation for $\alpha$ and $\beta$ from the Introduction. We now provide the intuition in **Section 5** and defer the formal derivation of the sampling utility to **Appendix A.2**.
>
> **Q5: The "Generation as a token-level MDP" section seemed to have inconsistent notation and questionable value.**
> **A5:** We have streamlined the Preliminaries. As detailed in **Overall Response Point 3**, we fixed the notation (standardizing $P$ for transitions) and moved the complex meta-MDP formalization to **Appendix A.1**. This ensures the preliminaries only cover necessary background without confusing variable reuse, while the rigorous formalism is preserved in the appendix for reference.
>
> **Q6: The Related Work section was lacking regarding inference-time control.**
> **A6:** We have expanded **Section 2** to include a broader range of inference-time control methods, including recent work on sequential sampling and input-adaptive consistency. We explicitly contrast these with ZIP-RC’s unique position as a single-pass, joint reward-cost approach.
>
> **Q7: The experiments lacked comparisons to the latest inference-time approaches.**
> **A7:** We have strengthened our baselines. We added **ZIP-RC-Lite** (frozen backbone) as described in **Overall Response Point 4**. We also emphasize our comparison against **Weighted BoN with self-evaluation (GenRM)** in Section 6, which represents a strong, contemporary inference-time baseline operating under similar compute constraints.
>
> **Q8: There was a broken figure reference and a typo regarding the logit slice.**
> **A8:** These are fixed. The broken reference is now the new **Figure 1** (see **Overall Response Point 1**), and we corrected the text in Section 4 to correctly refer to the slice over the reserved vocabulary $\mathcal{R}$.

---

### Author Response · Authors · 2025-11-26
**Overall Author Response to Reviews**

We thank all reviewers for their thoughtful feedback and constructive suggestions. We are encouraged that the reviewers recognize the **novelty** of repurposing reserved logits for zero-overhead introspection (Reviewers 5iv6, VsZ8, 5e4c), the **promise** of joint reward-cost prediction for adaptive inference (Reviewers hUjT, 5iv6, VsZ8, 5e4c), and the **strong empirical gains**, improving accuracy by up to 12% over majority voting and tracing superior Pareto frontiers (All Reviewers).

To address the primary concerns regarding clarity, implementation details, and generalization, we have significantly revised the paper. We have added comprehensive visualizations of the mechanism and its predictions, restructured the methods to provide formal derivations and clarify "zero-overhead" inference, implemented a new frozen-backbone baseline (ZIP-RC-Lite), and quantified distribution shift. Below, we summarize the major updates in the revision.

**1. New Comprehensive Overview and Visualization of Predictions (Figures 1 & 2)**
To address concerns regarding clarity (Reviewers hUjT, 5e4c), we added a new **Figure 1** that visually summarizes the end-to-end method: from repurposing the 8×8 grid of reserved logits to the resulting interpretable predictions and adaptive sampling utility.
To demonstrate the accuracy of these predictions (Reviewer hUjT), we added **Figure 2** and **Table 1**. These visualize the predicted joint distributions as heatmaps alongside the ground-truth distributions (estimated via 256 rollouts), quantitatively confirming via Total Variation that ZIP-RC accurately forecasts the outcomes of its own generation.

**2. Clarification of "Zero-Overhead" and Training Cost**
We have clarified the definition of "zero-overhead" throughout the text (Abstract, Intro, Section 4). We define it strictly in terms of **inference deployment**: ZIP-RC introduces *no additional model parameters* and requires *no additional forward passes*.
* **Training:** We clarify that ZIP-RC requires a one-time training phase (using ~100k labeled rollouts), which incurs a cost comparable to training standard Verifiers/Reward Models used in our baselines (Reviewer VsZ8).
* **Controller Compute:** We clarify that the sampling utility calculation (Section 5) is a lightweight CPU-side operation over scalar values, which is negligible compared to the GPU cost of the LLM forward pass (Reviewer 5e4c).

**3. Improved Presentation of Methods and New Formal Derivations**
We have restructured the paper to clearly distinguish between the prediction mechanism and the sampling strategy. **Section 4** now focuses exclusively on the ZIP/ZIP-RC mechanism (logits, grid mapping, training). **Section 5** now provides a concise, high-level overview of ZIP-RC Sampling and the sampling utility. In **Appendix A.1**, we formalize parallel sampling as decision-making under a meta-MDP. In **Appendix A.2**, this formalism allows us to derive the sampling utility as the value of an optimal predefined strategy that balances expected reward against the computational cost of extending prefixes.

**4. New Baseline: ZIP-RC-Lite (Frozen Backbone)**
To isolate the effect of updating the model backbone and to address the request for a probe-style baseline (Reviewers 5e4c, 5iv6), we added **ZIP-RC-Lite** in **Appendix A.5**. In this variant, we freeze the transformer backbone and train *only* the output head (reserved logits).
* **Results:** Figure 5 and Table 3 show that while ZIP-RC-Lite is less calibrated than the full ZIP-RC, it still enables effective pruning in latency-bound settings. However, because it overestimates reward variance, it is less efficient in compute-bound settings. This confirms that while the mechanism works as a probe, fine-tuning the backbone (with our KL constraint) yields superior calibration and performance.

**5. Quantitative Distribution Shift Analysis**
To address concerns about the impact of training on the base policy (Reviewers 5iv6, VsZ8), we added an analysis of the KL divergence between the ZIP-RC policy and the original frozen policy. **Figure 4 (Appendix A.3)** shows that with our regularization ($\alpha_{KL} = 10$), the KL divergence remains negligible (~0.005) throughout training. In contrast, removing the regularizer causes divergence to exceed 1.0, confirming that our objective effectively preserves the original generative capabilities.

---

### Meta-Review · Area_Chair_CBRL · 2026-01-09

**Summary:**

This paper introduces ZIP-RC, a “zero-overhead” inference-time mechanism that equips LLMs with real-time predictions of reward and remaining generation length by repurposing a small reserved subset of vocabulary logits in the same forward pass as next-token prediction.

Overall, the reviews were largely positive and converged on the view that leveraging reserved/unused logits for auxiliary prediction is novel and elegant. Reviewer 5iv6 appreciated the empirical results relative to the baselines, arguing that ZIP-RC outperforms existing approaches under the same cost. VsZ8 also highlighted the reserved-logit mechanism as an elegant idea and found the motivation compelling. This point that fixed-budget Best-of-$n$ approaches are wasteful because they generate all samples to completion regardless of marginal benefit resonated with the reviewer (and with me). 5e4c also viewed the approach of leveraging unused logits as interesting and innovative.

The main concerns in the original reviews centered on presentation and clarity. The mot negative reviewer was hUjT, who found the paper difficult to read and the method presentation opaque. Additional concerns raised by 5iv6 and VsZ8 related to distribution shift and the practical impact of the KL regularization term. For example, 5iv6 noted that the approach *“may still introduce distribution shift to some extent,”* and VsZ8 wrote that *“The KL penalty term $\alpha$KL is mentioned but its practical effect isn’t analyzed.”*

In their rebuttal, the authors made substantial revisions, including adding new figures to visually demonstrate prediction accuracy against ground truth, introducing an additional baseline, and adding a distribution-shift analysis. In my view, these changes address most of the major issues raised in the reviews, and I recommend acceptance. However, I strongly encourage the authors to make one more careful pass over the manuscript for clarity in the final version. This is an interesting method, and being crystal clear about how it works and its benefits can only increase the impact of the paper.

**Reviewer Concerns:**

Several key concerns were addressed by the rebuttal and revision. First, on clarity and opacity of presentation (raised most strongly by hUjT, and also by 5e4c), the authors added new figures and restructures the exposition to make the end-to-end method and the accuracy of the auxiliary predictions easier to follow. Second, in response to requests for stronger baselines (particularly from 5e4c), the authors added a new baseline (ZIP-RC-Lite). Third, regarding distribution shift and the KL term (raised by 5iv6 and VsZ8), the authors added an analysis that quantifies policy drift and demonstrates that their regularization keeps divergence small in their training setup.

**Reviewer Scores:**

Because reviewers mostly did not engage in the discussion, score-change estimates are my own.

5iv6 (8) was already strongly positive and would likely maintain their score.

VsZ8 (6) and 5e4c (6) were both marginally positive but flagged specific clarity, KL-effect, and baseline gaps. Given the authors’ added figures, baseline, and KL/distribution-shift analysis, I expect both would either maintain their overall stance or could likely increase slightly.

hUjT (2) raised primarily clarity and presentation issues. Since the rebuttal focused heavily on addressing these with new figures and reorganized exposition, I am inclined to believe hUjT would increase their score.

---

### Decision · Program_Chairs · 2026-01-26

Accept (Poster)